# Competition between physical search and a weak-to-strong transition rate-limits kinesin binding times

Trini Nguyen[1], Babu Reddy Janakaloti Narayanareddy[2], Steven P. Gross[2,3]*, Christopher E. Miles[4]*

1 Center for Complex Biological Systems, University of California, Irvine, Irvine, California, United States of America, 2 Department of Developmental and Cell Biology, University of California, Irvine, Irvine, California, United States of America, 3 Department of Physics, University of California, Irvine, Irvine, California, United States of America, 4 Department of Mathematics, University of California, Irvine, Irvine, California, United States of America

☯ These authors contributed equally to this work.
* sgross@uci.edu (SPG); chris.miles@uci.edu (CEM)

**Data Availability Statement:** MATLAB code to reproduce our results (compatible with version R2020a) is available at https://github.com/trininguyen/MotorBinding.

## Abstract

The self-organization of cells relies on the profound complexity of protein-protein interactions. Challenges in directly observing these events have hindered progress toward understanding their diverse behaviors. One notable example is the interaction between molecular motors and cytoskeletal systems that combine to perform a variety of cellular functions. In this work, we leverage theory and experiments to identify and quantify the rate-limiting mechanism of the initial association between a cargo-bound kinesin motor and a microtubule track. Recent advances in optical tweezers provide binding times for several lengths of kinesin motors trapped at varying distances from a microtubule, empowering the investigation of competing models. We first explore a diffusion-limited model of binding. Through Brownian dynamics simulations and simulation-based inference, we find this simple diffusion model fails to explain the experimental binding times, but an extended model that accounts for the ADP state of the molecular motor agrees closely with the data, even under the scrutiny of penalizing for additional model complexity. We provide quantification of both kinetic rates and biophysical parameters underlying the proposed binding process. Our model suggests that a typical binding event is limited by ADP state rather than physical search. Lastly, we predict how these association rates can be modulated in distinct ways through variation of environmental concentrations and physical properties.

## Author summary

Cytoskeletal-motor assemblies self-organize to achieve cellular functions ranging from delivering intracellular cargoes to generating forces in mitosis. Advancements in single-molecule experiments have revealed immense detail about motor detachment and stepping, but relatively little regarding attachment. Newly available binding times for individual kinesin motors allow for the evaluation of mechanistic models of the process. We find

**Funding:** This work was supported by the National Science Foundation Graduate Research Fellowship under Grant No. DGE-1839285 to TN and the National Institutes of Health under Grant R01 GM123068 to SPG. The funders had no role in study design, data collection and analysis, decision to publish, or preparation of the manuscript.

**Competing interests:** The authors have declared that no competing interests exist.

that a model limited by both diffusive search and a weak-to-strong transition from ADP release best explains the data. The coupled chemo-mechanical nature of this interaction can be modulated more richly than either separately, possibly explaining the diversity and regulation observed in cells. More broadly, our study provides a timely vignette in leveraging computations with experiments to understand the mosaic of mechanisms underlying protein-protein interactions.

## Introduction

Life depends on an immensely diverse and complex array of protein-protein interactions [1]. These interactions are richly regulated in both space and time (e.g., via post-translational modifications, fluctuating concentrations [2]) to modulate affinities, promiscuities, and sensitivities [3]. Understanding how these interactions are parameterized by both chemical and physical factors is broadly limited due to challenges in observing interaction events directly [4]. While predicting interactions from molecular structures (e.g., from molecular dynamics simulations) is an invaluable approach, these investigations still suffer from the same observational limitation in their validation [5].

One variety of such interactions of major importance across cellular function are those between molecular motors and cytoskeletal filaments. Cytoskeletal motors, specifically kinesin-microtubule assemblies, self-organize to perform a zoo of cellular behaviors, including the delivery of cargoes in intracellular transport [6], generation of forces to guide genetic material in mitosis [7, 8], and guiding of axonal growth [9]. Each of these wildly different behaviors is fundamentally achieved through molecular motors binding, stepping, and unbinding from cytoskeletal filaments [10]. Over the last several decades, advancements in single-molecule experiments have revealed extensive details about the latter two components [11–13]. Stepping and unbinding are, in some sense, *downstream* of binding, suggesting clear merit in understanding the details underlying the process.

Pursuits toward understanding motor-cytoskeleton binding have been clouded by complications in disentangling the measurements from convolving factors. That is, one must specify exactly the notion of binding that is being measured. To do so, consider the full process of self-assembly. Initially, a freely diffusing motor associates with a cargo, then the motor-cargo complex diffuses into close proximity to a cytoskeletal filament where a motor binds to the filament and begins stepping along it. Due to the challenges in disentangling these components of cargo-motor self-assembly, there is enormous variety in the reported ranges for motor binding rates. While landing rate assays [14] provide direct measurements of motor-cargo association rates, these do not inform motor-cytoskeletal rates. With the exception of [15], very little data of direct measurements of motor-microtubule binding events exists, but this study corresponds to the reattachment of a secondary motor that is kept close to the filament by another. Effective binding on the timescale of seconds [16, 17] to tenths of a second [18] have been reported from indirect measurements and utilized heavily in other modeling works [19–22] to understand collective motor behavior. However, these effective rates neglect geometric factors (such as organization on the cargo) that are known to crucially dictate the binding rate [23–26]. A mechanistic, biophysical model of the binding process is therefore necessary to reconcile the various experimental observations and modeling efforts.

Here, we use a combined experimental and computational approach to explore different possible biophysical models of how the motor-microtubule binding process occurs. The investigation is based on the initial association time between a cargo-bound kinesin and

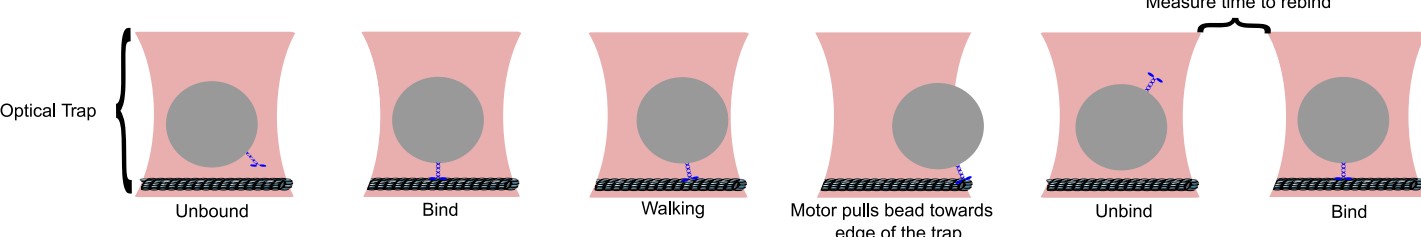

**Fig 1. Experimental setup.** An optical trap (pink) controls the average *z*-position of a polystyrene microbead cargo. When the cargo-motor ensemble binds to the microtubule and begins to walk on it, a position-sensitive diode (PSD) senses the displacement of the bead. As the motor walks farther from the center of the trap, the force on the cargo (and consequently the motor) grows, eventually leading to unbinding from the microtubule and resetting of the setup. The PSD measurements provide the timing between unbinding to rebinding, the binding process modeled throughout the remainder of the work. Further details can be found in [27].

microtubule from recent optical trap measurements depicted in Fig 1 on a variety of motor lengths and trapped distances away from the microtubule. The span of setups allows exploration and validation of models otherwise impossible with a single dataset. We first investigate the null model of a diffusion-limited search performed by the motor head. Through Brownian dynamics simulations coupled with simulation-based inference, we find that this model fails to capture a delay in binding at close distances. We find reconciliation with the data after the addition of an ADP-release requirement prior to binding to the model, motivated by known mechano-chemistry of motors. Through approximate Bayesian computation techniques, we quantify underlying rates and biophysical parameters governing this process and predict that most motor binding events are limited by tubulin-stimulated ADP release. Lastly, we provide predictions on how this process can be modulated distinctly by varying environmental concentrations or spatial distance, highlighting the complexity and regulatability of this interaction. Altogether, our study provides a new state-of-the-art mechanistic understanding of the motor-cytoskeletal binding process, a crucial ingredient in understanding the self-organization of motor-cytoskeletal assemblies used in cellular function. More broadly, our work illustrates how complexities arising from spatial and chemo-mechanical factors that shape protein-protein interactions may be understood through the combined efforts of theory and experiments.

## Results

### Diffusion-limited binding does not capture the qualitative behavior of experimental data

To investigate the biophysical mechanisms of the first association between a cargo-bound motor and a microtubule, we compare binding time data of three kinesin lengths (33, 45, and 60 nm) attached to a polystyrene bead that is laser-trapped at several distances away (0, 20, 40, and 60 nm) from a microtubule. Concentrations of motors in solution are diluted such that at most one motor is on each bead. Throughout the remainder of the work, we consider the binding time to be that between the unbinding reset event and the next time of detectable motion of the bead, as schematically shown in Fig 1. More details on the experimental setup can be found in Methods and in a concurrent manuscript with further validation [27]. The binding times for all setups are on the order of seconds, which is in line with other measurements of binding as discussed in the Introduction. Intuitively, as the cargo is moved away from the microtubule track, binding times increase. The most straightforward explanation for this is a "random search" mechanism rate-limiting the binding, schematically shown in Fig 2 as the "diffusion model". That is, the "null" model for binding, as assumed elsewhere [10], is that the

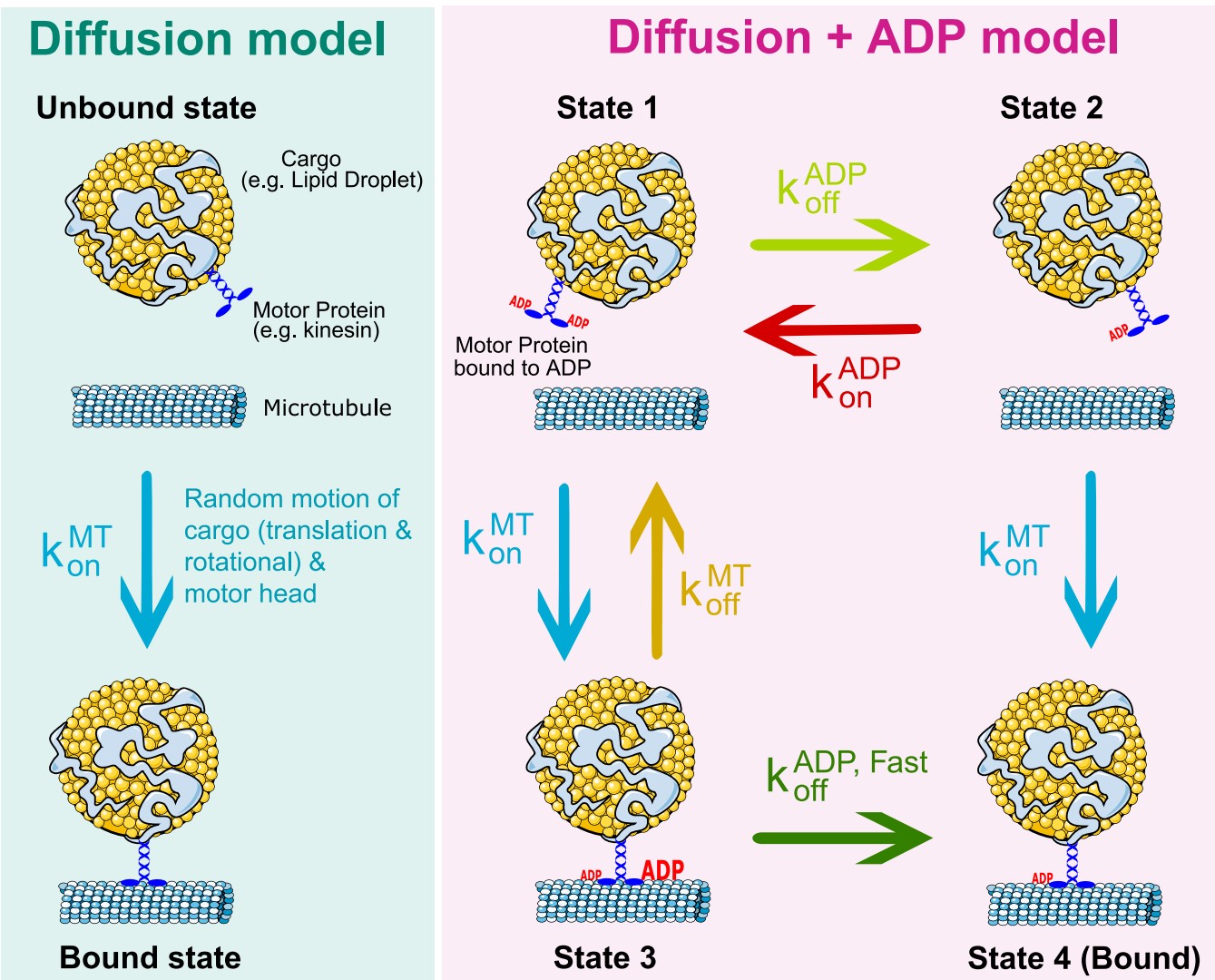

**Fig 2. Schematic of models.** Left: State diagram of the simple diffusion model of kinesin-microtubule binding. Through random motion of the cargo and motor head, binding time is determined by the stochastic search process of the motor reacting when in close proximity to the microtubule. Right: Model of binding process that considers a weak-to-strong transition driven by ADP release. We consider a cargo-motor ensemble that is in State 1, unbound from the microtubule and bound to ADP on both motor heads. From here, the motor can release ADP from one of its heads, transitioning to State 2, or bind weakly to the microtubule in State 3. To bind strongly to the microtubule and transition to State 4, the motor must meet two requirements: ADP is released from one of its motor heads and it must be within a binding distance to the microtubule. We consider two types of ADP release: a non-tubulin stimulated rate ($k_{off}^{ADP}$), and a faster tubulin-stimulated rate ($k_{off}^{ADP,Fast}$).

motor head undergoes random motion until it reaches close proximity to the microtubule track and then binds with some reactivity. Our work does not directly incorporate electrostatic effects known to underlie the binding process [28], but assumes that these effects can be lumped into the effective reactivity and movement parameters.

To investigate whether such a diffusion-binding model can explain the binding time across experimentally observed conditions (data shown in Fig 3), we developed a Brownian dynamics simulation of the proposed model. The stochastic model includes the random motion of the cargo, both translation and rotational, and the diffusive search of a motor head attached via a tether to this cargo. The tether is assumed to be of the known length of each motor and exerts a

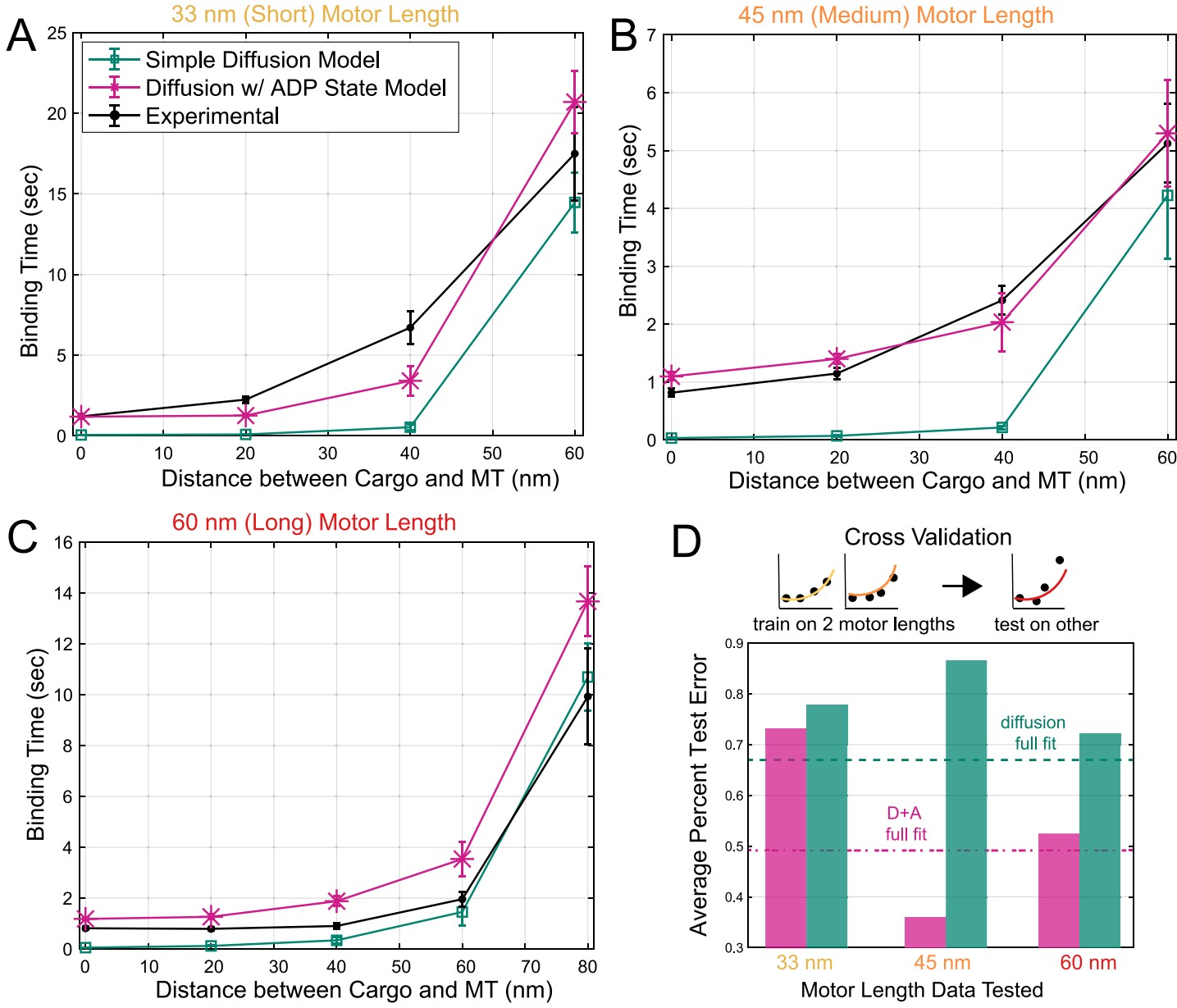

**Fig 3. Model fits and cross-validation.** ADP-release model captures qualitative behavior in experimental data, while the simple diffusion model cannot. *In vitro* optical trap experiments [27] were used to measure mean binding times (black) for three motor lengths: 33 nm (**A**), 45 nm (**B**), and 60 nm (**C**). The horizontal axis shows average distances between the cargo and microtubule (MT), which were varied for each experiment. Two binding models (simple diffusion only in green and ADP-release in pink) were simulated and fitted to all of the experimental data. *n* = 100 for simulated data varied for experimental data. Data are presented as mean ± SEM. **D**: Cross-validation was performed to determine the predictive power of each model. For three rounds, data was trained on two motor lengths, and tested on the third. Dashed lines show error when fitting the models to the entire data in panels **A-C**.

Hookean force when extended beyond this length. The initial configuration of the motor head is assumed to be downward, based on the fast timescale of resetting in the optical trap (tenths of a second). When the motor head enters a specified distance of the microtubule due to random motion, the binding reaction occurs at an unknown, microscopic rate. Additional model details and discussion of assumptions can be found in the Methods. Ultimately, this leads to two unknown parameters: the diffusivity of the motor head, and the microscopic reactivity.

**Table 1. Fitted kinetic and biophysical parameters.**

| Parameter | Simple Diffusion Model | Diffusion w/ ADP State Model | Previous Literature |
|---|---|---|---|
| $k_{\text{off}}^{\text{ADP}}\,(\text{s}^{-1})$ | 0 | 0.008 | 0.008–0.1 [34–36] |
| $k_{\text{off}}^{\text{ADP,fast}}\,(\text{s}^{-1})$ | 0 | 2.12 | 0.5–300 [35–41]* |
| $k_{\text{on}}^{\text{ADP}}\,(\text{s}^{-1})$ | 0 | 883.9 | 425.0 [35] |
| $k_{\text{on}}^{\text{MT}}\,(\text{s}^{-1})$ | 80.6 | 70.65 | |
| $k_{\text{off}}^{\text{MT}}\,(\text{s}^{-1})$ | 0 | 0.2 | 0.31 [42] |
| $D_{\text{m}}\,(\text{nm}^2\,\text{s}^{-1})$ | 4459.8 | 1994.0 | |
| $\kappa_{\text{w}}\,(\text{pN}\,\text{nm}^{-1})$ | 0 | 0.0020 | |

Parameters for both models fit using a Bayesian optimization algorithm [29] and compared to literature values when available.

*The wide range of ADP release rates can be attributed to force-sensitivity, as identified in [39]. See text for further discussion.

Through a suite of simulation-based inference techniques [29–31] (further details in Methods), we obtain fits to the diffusion model over all experimental setups for the two unknown parameters: the diffusivity of the motor head, and the microscopic binding rate. The resulting fits can be seen in Fig 3A–3C in green for the mean time to bind for the three motor lengths and various distances. There appears to be reasonable qualitative agreement with the experiments, where increasing distances increases binding time. The corresponding parameter fits can be found in Table 1. While the diffusivity of the motor head is challenging to quantify [32], our fitted value on the order of 1000–10000 nm²/s is within ranges considered for kinesin elsewhere [33]. Upon further scrutiny, the mean binding times shown in Fig 3A–3C, especially at close distances, display a distinct qualitative disagreement between the diffusion model and experimental measurements. In the diffusion model, as the cargo is trapped closer, the motor is effectively instantly able to bind. However, experimental values show a plateau of times around 1 second, even for close distances. This plateau points to the binding process being a multistep process.

## A chemo-mechanical ADP-release model of binding better explains observed binding times

With the observation that a simple diffusion model does not produce the $\sim 1$ second delay in binding at close distances seen in experimental measurements, we sought a model that may explain this phenomenon. Several plausible explanations including cargo rotation and measurement error were considered, but seem unlikely when evaluated with estimates of their effect (see Discussion). Instead, we turn to the rich mechanochemistry of the kinesin motor. It is known that the nucleotide state of each motor head crucially determines its strong or weak affinity to the microtubule [43–45] and through cycles of this nucleotide state (ATP, ADP, released), processive stepping is achieved [46]. We posit that this nucleotide-based regulation of "binding" extends beyond that of processive stepping, and even the preliminary attachment between the cargo-motor ensemble and the microtubule. That is, we posit that the experimentally observed binding times correspond to a strong binding event, and therefore the underlying nucleotide state of the motor heads, plays a significant role in arriving at this state.

To investigate whether a model including nucleotide state may better explain the experimentally observed times, we extend the model to account for 4 possible states, as shown in Fig 2. In this model, the motor-cargo ensemble begins in State 1 with both motor heads in an ADP state, undergoing the same random motion as the diffusion model. From here, the ensemble can enter one of two states: State 2, where ADP is released from either motor head,

or State 3, when the ensemble diffuses close to the microtubule and one of the motor heads weakly binds to it. From either of these states, the ensemble can then strongly bind to the microtubule either through diffusion (from State 2), or the ADP molecule is released (from State 3). We consider two types of ADP release, a fast tubulin-stimulated release and a slow non-tubulin-stimulated release [34–38]. We consider ADP release as a requirement for strong binding based on the neck-linker model for stepping where an ADP-bound head has a low affinity for the microtubule, then this trailing head moves forward along the microtubule bound to ADP, and when it steps down onto the microtubule, ADP is released [47]. Importantly, our description is coarse-grained to not track the heads separately, but we consider the ADP release to describe either motor head. The assumption that the ensemble begins in State 1 has two parts: we assume that if the motor detaches in an ATP-bound state, this phosphate release is fast [46], but then the corresponding ADP release is slower without tubulin [37].

Using the same simulation-based inference approaches for the diffusion model, we fit the observed binding times for all 3 motor lengths and distances simultaneously for the extended ADP-diffusion model, with 7 unknown parameters, 2 from the diffusion model, 4 reaction rates, and 1 corresponding to the strength of attachment in the weak binding state. The result of the fits can be seen in Fig 3A–3C in pink. The overall fit is discernibly better for the 33 nm and 45 nm motor lengths, and arguably worse for 60 nm at long distances. The noteworthy consistent overestimation for 60 nm motor arises due to the simultaneous fitting of fixed parameters across all three motor lengths, sacrificing better fits for shorter motor lengths at the expense of the longest length. Such consistent overestimation does point toward a shortcoming of the model and may be due to heterogeneities in the different motors beyond their length alone. However, the model now importantly captures the qualitative feature of a plateau of times at short distances. While only the mean binding times were used to fit, Figs B and C in S1 Text show close agreement in full distributions of binding times as well.

Beyond the qualitative improvement, the inherent danger in quantitatively assessing whether the ADP-release model better explains the data comes from the increased model complexity [48]. Intuitively, a model with more parameters has more flexibility to produce a better fit, and careful attention must be paid to model selection. In lieu of commonly-used information-theoretic techniques (AIC, BIC), even for simulation-based inference [49], we instead leverage the structure of our experimental observations to compare models based on their ability to explain unseen experimental circumstances. We perform a cross-validation procedure where we fit both the diffusion and ADP+diffusion models to the binding times for 2 of the 3 motor lengths, withholding one for testing on the trained models. In each validation test of withholding a motor length, the more complex model generalized better, shown in Fig 3D. From this, we conclude that the ADP-diffusion model better fits the observed binding times, even under cross-validation-based scrutiny [50].

## Kinetic and biophysical parameters of the ADP-binding model can be estimated with high precision

Beyond the qualitative lesson of identifying the ADP-diffusion model as explaining the data, our fitting procedure provides rich quantitative insight into the underlying processes by estimating underlying parameters, shown in Table 1 and Fig 4. Some kinetic rates have been previously measured, and serve as support for the model, whereas others are, as far as we know, unmeasured. The values reported in Table 1 are point estimates from a simulation-based inference optimization procedure [29]. The estimated values for microscopic binding rate, diffusion of the motor head, and ADP binding are all within an order of magnitude of previously reported estimates. To our knowledge, the weak-tethering strength $\kappa_w$ has not been reported

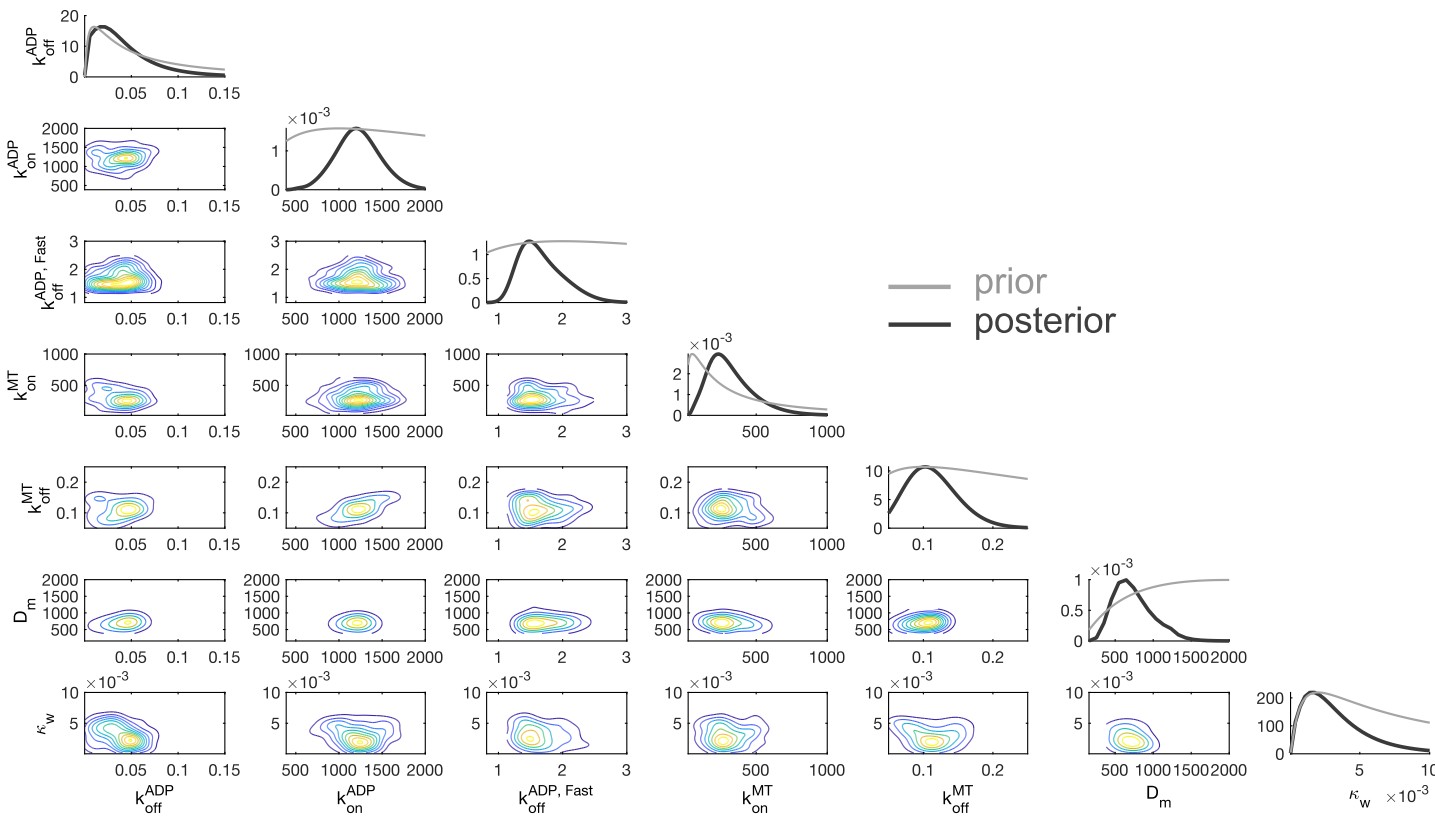

**Fig 4. Estimates of microscopic parameters from fitting the ADP-diffusion model.** Joint (Approximate) Posterior Distribution of ADP release Model Parameters. Black curves in the marginal densities from sequential approximate Bayesian computation (sABC) are the posterior distributions, and the grey curves are the priors. A kernel density estimator [51] was applied to discrete samples to form posterior estimate curves.

elsewhere, but we note it is significantly weaker than other physical forces in the system and may correspond to electrostatic attraction. The limiting ADP release rate estimated by our model is ≈ 2/s. This parameter has a wide range of values reported in the literature, ranging from slow rates in the vicinity of ours, as well as significantly faster rates on the order of hundreds per second. We defer discussion of this important parameter and its subtle interpretation to the Discussion. As further validation of this optimization procedure for point estimates, we also performed a separate simulation-based inference technique, sequential approximate Bayesian computation (sABC) [30] to obtain samples of an (approximate) posterior distribution shown in Fig 4. The reason for this method was two-fold: for one, the agreement between the point estimates arising from the two procedures validates the approximations involved in the techniques, and the latter sABC approach produces valuable uncertainty quantification that we were unable to employ but may very well be possible using the techniques of [29]. Further details on these procedures can be found in the Methods. Somewhat surprisingly, all parameters of the model seem to be identifiable, as shown in the relatively tightly-shaped posterior distributions.

## Most motors strongly bind via tubulin-stimulated ADP release

These quantitative estimates of the underlying microscopic rates provide qualitative lessons about motor binding. Referring to the model schematic in Fig 2, motors can achieve the strong

binding state either through an intermediate weak binding state (State 3) after which ADP release occurs, or directly from a diffusing state (State 2). Although we do not observe these transitions directly, their relative proportions can be deduced from the data and are investigated in Fig 5. Our proposed binding process can be conceptually decomposed into two components: the physical search process of the motor binding (either weakly or strongly) to the microtubule, and an ADP release step. Fig 5A demonstrates the predicted relative contribution of the physical search component to the overall binding time. Specifically, the panel shows the fraction of time stochastic simulations of the full binding process spent in States 1 and 2, where the motor is unbound completely and searching for the microtubule. This fraction was calculated by determining the portions of time spent unbound in each simulation, and then taking the average. For all motor lengths, the fraction of time in the searching state increases as

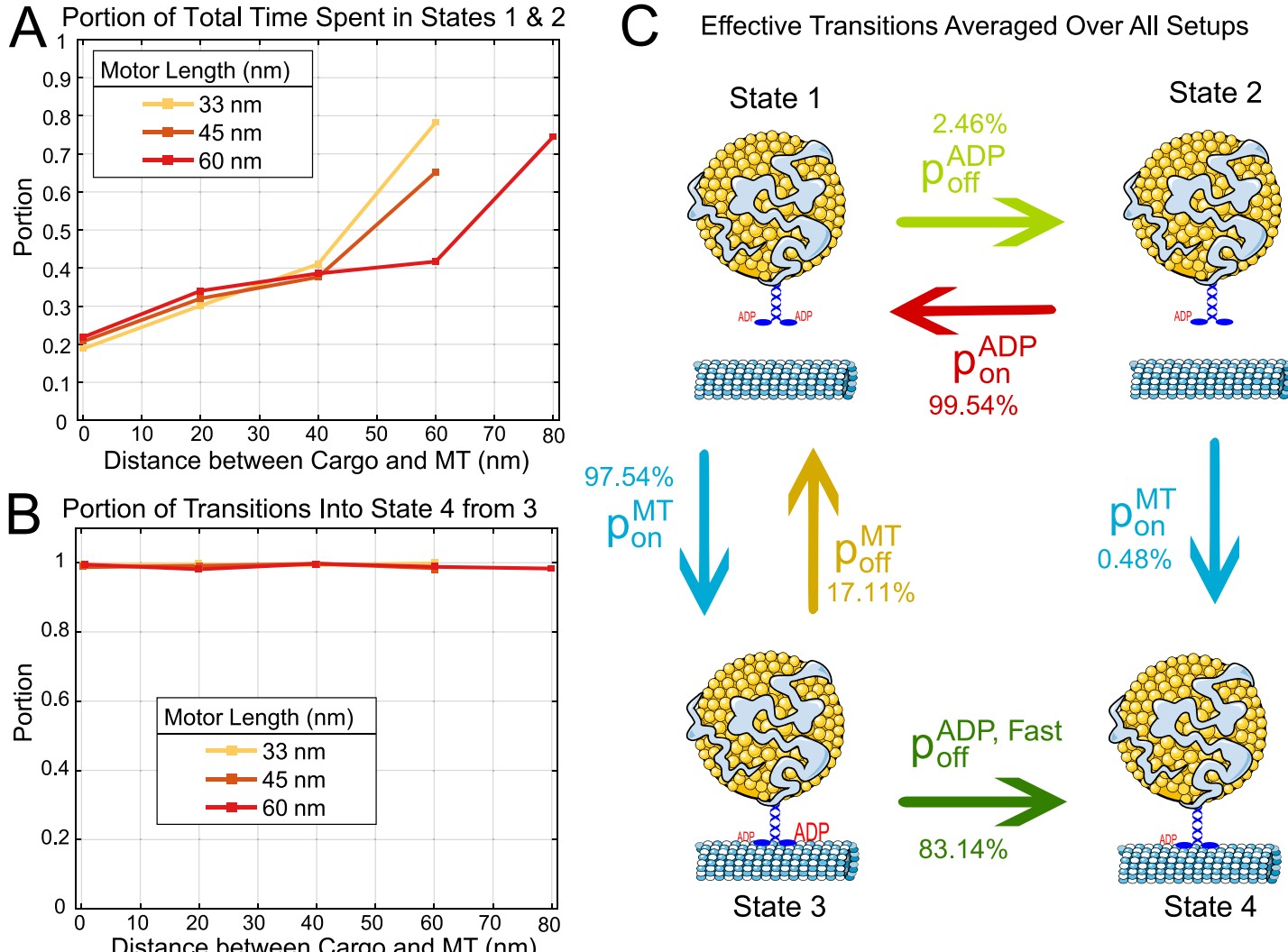

**Fig 5. Cataloging pathways of binding from stochastic simulation shows that motors typically bind via tubulin-stimulated ADP release. A:** Averaged portion of total time spent in the unbound states (1 and 2). For all motor lengths and distances, this portion increases. **B:** Portion of all binding events that weakly and then strongly bind (arrive in state 4 from state 3) from state 3 (as opposed to arriving from state 2). **C:** Mean probability of transitions that occur from each state, averaged over all events from all experimental conditions (motor length and mean spacing between cargo and microtubule). Parameters for simulation are from Table 1.

the distance between the cargo and microtubule increases, ranging from about 20% to 80% of full binding time. Moreover, for a fixed distance (60 nm), the fraction of time searching increases as the length of the motor decreases. These trends can be interpreted as the diffusion-based physical search step always contributing a meaningful rate limitation to the process. However, from this panel alone, the typical binding pathway cannot be deduced. As physical search takes varying of the portion of the total binding time, is ADP release always, never, or sometimes the rate-limiting component? Fig 5B addresses this question by computing the fraction of stochastic simulations that end up strongly bound (State 4) by entering through the tubulin-stimulated ADP release pathway (from State 3). These portions show that for all motor lengths and distances, effectively all binding events (about 99%) enter through this state.

Motors could, in theory, release ADP while undergoing diffusive search, and then strongly bind directly (States 2 to 4). To understand why this pathway does not seem to contribute to the binding time, we show the full effective predicted binding transition frequencies in Fig 5C, averaged over all motor lengths and distances. A full report of the relative proportions and effective rates of each transition can be found in Figs E and F in S1 Text. From State 1, most (over 80%) of initial binding interactions arise via a preliminary weak binding state and subsequent ADP release (States 3 to 4). However, we predict that some transitions to State 2 (ADP release while unbound) occur. From State 2, there may seem to be a paradox that the transition rate between States 2 to 4 is very low, which one might interpret as the ADP released motor state having *low* affinity for the microtubule, but we emphasize this quantity is a byproduct of the competing rates between physical binding and ADP capture. That is, State 2 does not have a lower affinity for the microtubule, but rather, occurs less often than ADP rebinding. Therefore, the ADP release before physical search is too transient to provide a viable binding pathway. Altogether, our results suggest tubulin-stimulated ADP-release after weak binding is the typical pathway for kinesin binding.

## Binding rates can be distinctly modified by physical and chemical factors

The quantity and structure of experimental data have thus far allowed for significant progress in understanding binding from retrodictive reconciliation with a model. We conclude with predictions that emphasize the broader lessons and may serve as the basis of validation in future experiments. The exploration of typical binding events in the previous section points toward conceptualizing this process as a distinct mechanical diffusive search and a chemical step from the nucleotide state. Regulation and perturbations of each of these components should therefore be discernible. To explore these two scenarios, we predict how the binding times should be altered in two hypothetical experimental perturbations shown in Fig 6. Fig 6A shows how if one could modulate the tubulin-stimulated ADP-release rate, the effective spatially-dependent binding rate shifts up or downward for all distances. In contrast, numerous physical experiments could plausibly alter the physical properties of the system. In Fig 6B, we show the predicted effect of changing the cargo size, which would consequently modify the overall diffusivity of the ensemble and the random search time. For short distances that are not limited by this diffusive search, the difference is negligible, but for long distances, the effect becomes magnified. Predictions for other motor lengths can be found in Fig G in S1 Text. While we do not currently have the technological capability to validate these experimental predictions (e.g., the ability to disentangle the competition between ADP and ATP binding to the motor head prior to the motor binding), we hope they will be the basis of future validation or invalidation of our proposed binding model.

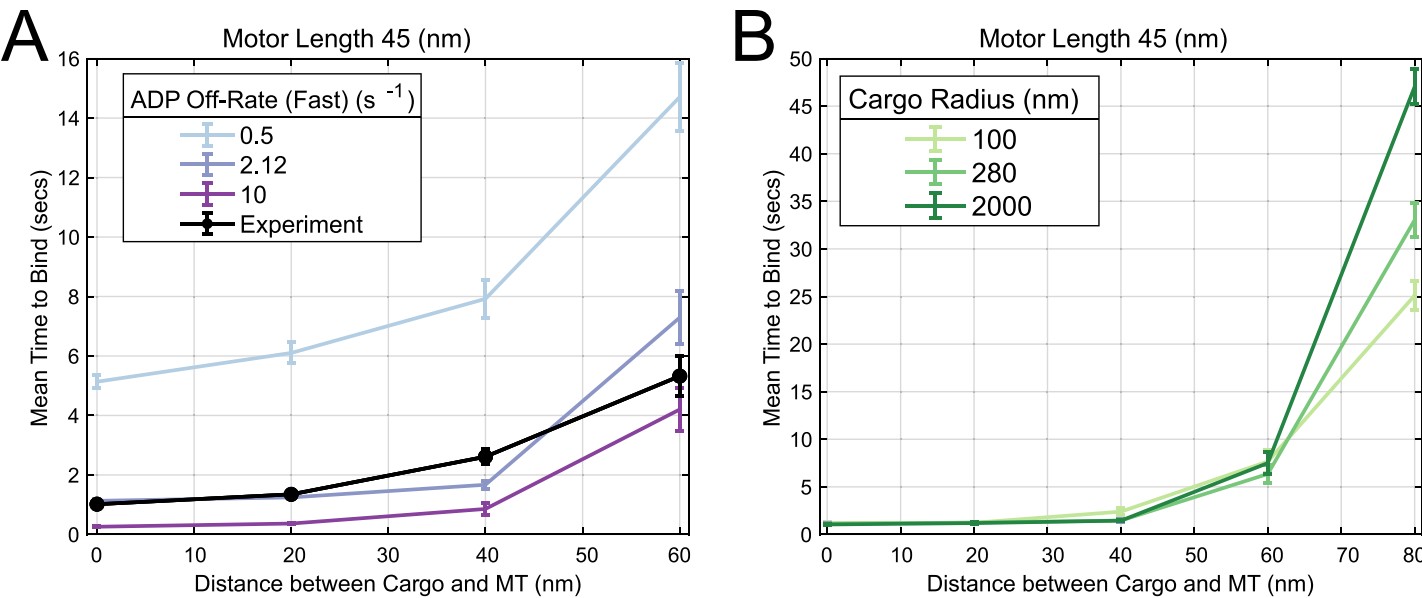

**Fig 6. Predicted binding time changes from chemical and physical perturbations. A:** An example of a chemical change, $k_{\mathrm{off}}^{\mathrm{ADP,Fast}}$, was varied in the simulation and resulting simulated binding times are plotted. Data are presented as mean ± SEM. **B:** Physical changes, such as changing the cargo size, can also be made to study the resulting binding times.

## Discussion

### Conclusion

Altogether, our results point toward a model of the initial binding between a cargo-bound kinesin and a microtubule track being more complex than a diffusion-limited search of the motor head that is presumed elsewhere [10, 33]. Motivated by the known mechano-chemistry of motor stepping, where the nucleotide state of each motor head dictates microtubule affinity [34, 37, 52], we posit that the observed binding times correspond to a nucleotide-state-limited strong binding event. In this conceptual model, the primary binding pathway is a preliminary weak binding from physical search, followed by the motor being weakly tethered to the microtubule. During this weak tethering, ADP is released from one of the heads and the motor becomes strongly bound. The rate-limiting component transitions between ADP release and physical search as the cargo-microtubule distance increases, arising from the competition between these rates. To validate this hypothesis, we considered a coarse-grained computational model that incorporates both diffusive search and the ADP state of motor heads, and through simulation-based inference and model selection, ultimately found compelling agreement with the experimental measurements. With only observed binding times, computational modeling reveals unobserved details of the binding process and predicts that the "typical" binding event is modulated distinctly by both environmental and physical factors. From a design perspective, these orthogonal modulations allow for more fine-grained control and malleability than either separately.

The emergent model of an ADP-release rate-limiting kinesin binding warrants further discussion. Due to the inability to observe the behavior of individual motor heads, our computational model forgoes this complexity and implements weak and strong binding states agnostic to the underlying biochemistry. The simulation-based inference procedure and its validation rigorously show that a model with this weak-to-strong transition fits the binding data more

 

closely than without these states. This agreement primarily arises from the fitted $\approx$2/s rate of strong binding after weak binding. Based on the known mechanochemistry of kinesin, we attribute this weak-to-strong transition to an ADP release of one of the motor heads [34, 37, 52]. However, this rate is commonly reported on the order of $\approx$100/s [40], allowing for the rapid procession of the motor on this same rate-limiting timescale. This presents an apparent contradiction to our hypothesis, as this fast rate indeed fails to explain the apparent delay seen in binding times. Reconciliation arises from the findings of [39], principally the result that ADP release rate of each kinesin head is highly sensitive to the load applied to it. When both kinesin heads are bound to the microtubule, the release rate for ADP from the front head (under load) is indeed very rapid. However, the ADP release rate for kinesin when only a single head is bound (and not under load) is much lower. Our model describes the transition from kinesin completely unbound to the MT to a single-head attachment and therefore corresponds to this slow unloaded, microtubule-stimulated ADP release rate. This slower ADP release rate has been reported in the literature with noteworthy agreement with our estimated rate. In [53], the authors find a bimodal distribution of ADP release rates: one fast, and a second, slower rate of 0.4–2.3/s. This is in close agreement with the rates of 2.3–3.3/s found in [54], although for a different motor (ncd). These same authors later report a faster value of 31.5/s [36], but note this is an average of fast and slow rates. In a more recent study [41], the authors find an ADP exchange rate of $\approx$0.5/s that can be modulated by engineering longer neck linkers. Admittedly, our proposed model seems in tension with the historical narrative that the slow ADP release corresponds to the *second* head after the first quickly releases ADP. A possible reconciliation arises from the idea that a single head binding event (with fast ADP release) is not sufficiently strong alone to prelude stepping, but the second, slower ADP release of the next head leads to the binding events observed in the experimental assay. However, like the studies leading to previous conceptual models, our work cannot resolve the precise underlying biochemistry of individual motor heads. Nonetheless, the remarkable agreement between the fitted weak-to-strong transition rate arising from our work and the slow unloaded ADP release found in the literature provides plausible evidence of their connection. Moreover, the emergence of the unloaded ADP rate reveals more insight into the underlying mechanics between the motor heads in the attachment process.

Motor binding times have been estimated and measured many times prior [14–17], but these studies cannot easily disentangle the physical configuration the binding arises from, whether it be from a landing experiment, DNA origami, or motors re-attaching while another is already bound. The absence of this consideration juxtaposes the increasing body of evidence that spatial organization plays a vital role in motor binding [23–26]. By distilling both the experimental assay and corresponding model to the minimal ingredients of a single motor attachment, we can clarify this process with unprecedented precision and generalizability. That is, while we have shown that our model successfully recapitulates the experimental data from our optical trap assay, we have moreover provided quantitative details about the underlying process that can be used to calculate binding times in other configurations. That is, one could imagine taking our fitted model and adapting it to DNA-origami cargo to reconcile the observed binding rates of [15, 17]. Discrepancies between the predictions and observations may occur, but these provide crucial details about the underlying chemistry and physics that we advocate warrant further investigation.

## Limitations and assumptions

We have not, and likely cannot ever fully rule out other conceptual models and confounding factors of binding time. The key qualitative feature we sought to replicate was the binding

delay on the order of 1 second in close-proximity cargoes. Our ADP release model successfully recapitulates this, and we provide testable hypotheses that can be used to invalidate the model in Fig 6. We considered several other possible factors that may explain or contribute to this delay. The most pressing possibility is whether this delay arises as an artifact of failure to detect optical trap displacements faster than this. Assuming an unloaded kinesin velocity of around 500 to 1000 nm/s, a 15 nm bead displacement for detection corresponds to fractions of a second and does not explain the observed delay on the order of a second. Furthermore, we struggle to speculate what a realistic initial configuration of the motor may be, as a more detailed model of the cargo resetting in the optical trap likely requires careful consideration of hydrodynamic effects that couple rotation, motion, and distant-dependent diffusivity [55]. We made the crude assumption that the motor was configured downward initially due to the fast timescale between cargo resetting in the trap, but this is likely not the case. However, our estimates (see Fig H in S1 Text)) and others estimate [26] a cargo taking 0.2 seconds to complete a full rotation at this viscosity. To test the robustness of this result, we extended the model to also approximately incorporate near-wall diffusivity hindrance from hydrodynamics [56–58] and find that there is some slowdown at far distances with random initial motor configurations, shown in Fig K in S1 Text. In a previous study from the lab [59], it was found that a reduced bead size does reduce the binding time slightly. Altogether, this evidence points toward supporting our conceptual model that cargo rotation and hydrodynamics do play a role in motor binding time, but alone fail to explain the magnitude of delay seen at close binding distances. One last possibility we note is the conceptual model where a motor begins in a "crumpled" state, and then unfurls with some delay to bind. Our model of the kinesin stalk is crude, and one could imagine other possibilities such as a worm-like chain model considered for the neck linker [60]. However, these models primarily differ when under load, rather than undergoing a diffusive search. Therefore, we leave the investigation of other polymer models for the stalk to future work. We note the coarse-grained approximation of the whole motor head as a single spatial point with a single nucleotide state due to the inability to resolve further detail from available data. We leave to future work more detailed models that incorporate *in vivo* complexities, including motor attachments diffusing on the surface of cargo [23–25] or cargoes with multiple motors. Lastly, we identify that this binding model may be limited to kinesin, and perhaps even only some kinesins. Future investigation warrants investigating other motors' binding details, e.g., dynein, through the procedure outlined in this work.

## Context in cytoskeletal-motor systems

The focus of this work is understanding the binding between a single cargo-bound kinesin and a microtubule track. This setup follows the spirit of a now long-established and successful line of investigation of cytoskeletal-motor assemblies by isolating fundamental building blocks. We attempt to situate our advances in the broader context of the wildly complex array of cytoskeletal-motor interactions and the feedback between them. We emphasize the chemo-mechanical nature of our binding model in the context of the enormous literature on how physical and chemical changes to microtubules affect motor behavior. For instance, geometry of the microtubule network dictates cargo-microtubule distances [61, 62] by pulling the cargo closer to the microtubule via tethers such as dynactin [63], or pushing it away via microtubule-associating proteins (MAPs) [64], and motors themselves reorganize microtubules through forces [33]. A zoo of MAPs and the tubulin code are known to interfere with motor function [65, 66], including in the recruitment of motors [67–69]. Our model may shed light on explaining the mechanisms by which these microtubule decorations modify motor binding, and serve as the basis for future investigation. Through the decomposition of how chemical and physical factors

modulate binding, our study may be the basis for discerning the mechanism of MAPs regulation of motors. For instance, one could imagine C-terminal tails may serve as physical tethers or may alter nucleotide states, and such an investigation remains for the future. More broadly, this understanding serves as a key step toward understanding how cells regulate binding to direct cargo and perform even more cytoskeletal-motor functions such as coordinate mitosis [70]. Moreover, this understanding may help aid in the design of increasingly sophisticated synthetic systems [71], where spatial distances can be prescribed.

### Broader lessons for probing protein-protein interactions

The difficulty in directly observing protein-protein interactions makes their study challenging. Two main avenues of approaches have been historically successful, each at extremes of chosen level of detail. Molecular dynamics simulations are a gold standard for predicting interactions. We build on the immense insight they have illuminated on the interactions between microtubules and kinesin [72, 73] and otherwise would not have considered the ADP release in our model. However, these approaches built from microscopic components have immense difficulty in scaling up to more complex systems with multiple interacting components, such as between motor, cargo, and cytoskeletal filaments. At the other extreme, "spherical cow" models of diffusion-limited reactions [74, 75] have revealed many qualitative lessons of protein-protein interactions, but remain challenging to quantitatively link with data because even the inclusion of modest complexities like orientation constraints [76] make the analyses prohibitively complex.

Our work adds a timely vignette to other studies [77–79] that illustrates the value of striking an intermediate level of complexity in understanding protein-protein interactions. This balance allows for the incorporation of microscopic details from more fine-grained studies but remains vigilantly coarse-grained to directly connect with data. We highlight major components of the work that we believe will be of broader use in other probing of protein-protein interactions, such as understanding the competition between peptides and kinases for the same binding site on a transcription factor [80], or disentangling folding and aggregation rates in proteins [81]. For one, we leveraged measuring interactions in a variety of conditions. Equipped with only a single motor length or trapped distance, the diffusion model would have fit to a seemingly satisfactory degree. By probing a model's ability to explain data across conditions, we were able to identify the ADP model. Furthermore, our work was made possible by recent advances in simulation-based inference [82]. While model fitting has historically been bogged down in the complications of the procedure, we now live in an age where it is plausible to perform inference on any model that can be thought of (and simulated), with rapidly improving techniques beyond those we utilized in this work [83]. Neither of these lessons is specific to the context of cytoskeletal-motor interactions, and therefore we hope our work serves as an outline for other pursuits in data-driven discovery of protein-protein interactions.

## Methods

### Optical trap experiments

The optical trapping setup was assembled on an inverted Nikon TE200 microscope using a 980 nm, single mode, fiber-coupled diode laser (EM4 Inc). The laser power was set to achieve a trap stiffness, $\kappa_t$, of $\sim 0.045$ pN nm$^{-1}$ while using the polystyrene bead of 0.56 $\mu$m (streptavidin conjugated, Spherotech).

Single motor experiments were carried out in the motility buffer (80 mM Pipes pH 6.9, 50 mM CH3COOK, 4 mM MgSO4, 1 mM DTT, 1 mm EGTA, 10 $\mu$M taxol, 1 mg mL$^{-1}$ casein). In all the rebinding rate assays, single-motor kinesin-coated polystyrene beads were prepared

just before the measurements. The motors DK-406-His/DK-560-His/DK-746-His (Kinesin-1, aa 1–560/Drosophila Kinesin aa 1–406/ Drosophila Kinesin aa 1–746; His tag at c-term) were diluted to ∼20 nM before mixing with ∼1 pM of biotinylated penta-His- antibody conjugated streptavidin beads stored at 4°C. This ratio produced the bead binding fraction of 10–15% and was maintained to maximize the probability of finding single motor beads in the solution (a bead binding fraction less than 30% corresponds to a single motor regime [84]). The bead-motor incubation (∼50 $\mu$L volume) was carried out at room temperature for 10 minutes. At the end of incubation, the sample chamber with preassembled microtubules was washed with ∼50 $\mu$L of warm filtered buffer just before injecting the incubated mixture. Experiments were carried out at room temperature in a motility buffer supplemented with 2 mM ATP and oxygen-scavenging system (0.25 mg mL$^{-1}$ glucose oxidase, 30 $\mu$g mL$^{-1}$ catalase, 4.5 mg mL$^{-1}$ glucose).

In general, small dust or debris in the solution gets pulled into the trap along with the bead. Trapped dust interferes with motor rebinding to microtubules. To prevent this interference, the large dust particles and aggregates of casein in the buffer were removed using a 100 nm centrifugal filter (Millipore). Another potential issue is the stage drift during measurement, and it was minimized with an automated drift correction system using an xyz piezo stage (PI) and custom software.

All kinesin proteins were purified using HIS-tag and MT affinity purification after expressing them in Rosetta bacterial cells as described earlier [85]. DK406 plasmid was procured from Addgene (plasmid ID #129764, generously supplied by William Hancock lab). DK746 was designed in the lab after modifying the full-length DK980, also procured from Addgene (plasmid ID: #129762, William Hancock lab), using restriction enzyme digestion.

**Binding detection.**   Bead displacements in the trap registered by a position-sensitive diode (PSD; First Sensor AG) were acquired at 3kHz using an analog-to-digital converter (ADC) card. The digitized PSD data was smoothed with a 40 point fast Fourier transform (FFT) filter and analyzed by custom Matlab code to score all the peaks greater than 15 nm and lasting more than 0.01 seconds (30 data points). The experimental method is fully described in [27].

**Maintaining bead-microtubule separation.**   Autocorrelated template matching of defocused fiduciary bead immobilized on the surface served as a feedback signal to maintain a stable bead-microtubule separation. The method was developed using an autocorrelation of a template image with real-time images of the fiduciary beads to generate a parameter called match score (degree of matching). It is custom-developed to study protein-protein interactions using an optical trap. To describe it briefly, fiduciary beads were immobilized on the coverslip and an image of the bead was recorded at 200 nm below the surface to serve as a reference library image. The key parameter here is the match score (value = 0 for no match and 1000 for perfect match with the library image) and when the template used is a bead in focus, this score exhibits quadratic behavior in the vicinity of the surface. Thus, score change per nm of the focus shift is negligible when the bead is in perfect focus. However, when a defocused bead image is used as a reference library image, the score change/nm is as high as 1% for every 10 nm z-focus change. This parameter was used as a feedback signal both to increment and lock the z-position at a fixed level using an automated focus locking system developed in-house using an xyz-piezo-stage, image grabber cards, and labVIEW. In the experiments, the distance between the trapped bead and the Surface-attached microtubule is altered by moving the surface. In principle, moving the surface by causes a slightly smaller change (≈12% less) in the distance between the trapped bead and the surface, due to the mismatch in refractive indices. This has not been corrected for in the data presented.

## Brownian dynamics simulation

The simulation consists of a motor that is bound to a cargo and a microtubule. The cargo is a three-dimensional sphere and is subject to translational and rotational diffusion. The motor's condition is dependent on whether an ADP molecule is bound to it and whether it is bound to the microtubule. Whether the motor is weakly or strongly bound to the microtubule is dependent on whether it is bound to an ADP molecule. The transitions through these states (Fig 2) are simulated using a Gillespie-style algorithm [86]. The motor is defined by its location of attachment to the cargo and its head location. Locations of the motor head and cargo center are calculated using the Euler-Maruyama method [87]. The microscopic binding between the motor head and microtubule follows a standard "Doi model" for chemical reactions [88]: when the motor heads come within binding reach of the microtubule, it has a constant rate of binding to it; otherwise, this rate is 0. The motor behaves as a spring, and when they are bound, they experience and exert force. When the motor is weakly bound to the microtubule, its off-rate depends on force. ADP molecules can also bind and unbind to the motor head at constant rates, but the ADP off-rate is dependent on whether the motor is weakly bound to the microtubule. The equations of motion for the cargo and motor are constructed by discretizing a set of stochastic ordinary differential equations derived from force balance.

**ADP release model description.** This model is three-dimensional and mesoscale, with state variables and summarized in Table 2. A set of stochastic ordinary differential equations is used to describe the location of the cargo sphere and the motor that is attached to it. The motor transitions stochastically between discrete states (shown in Fig 2), and these transitions occur as Poisson processes. The force that the motor exerts on the cargo is modeled as a one-way spring:

$$
\vec{F}^m\left(\vec{a}, \vec{h}\right) = \begin{cases} -\kappa^m \left(\left|\vec{h} - \vec{a}\right| - L_m\right)\left(\frac{\vec{h} - \vec{a}}{\left|\vec{h} - \vec{a}\right|}\right) + \vec{F}^w & \left|\vec{h} - \vec{a}\right| > L_m \\ 0 & \left|\vec{h} - \vec{a}\right| \leq L_m \end{cases},
$$

where $\vec{a}$ and $\vec{h}$ are the motor anchor and head locations, respectively, $\kappa^m$ is the motor stiffness

**Table 2. List of variables in the computational models.**

| Variable | Description |
| --- | --- |
| $\vec{F}^m$ | Force from motor head |
| $\vec{a}$ | Position on cargo where motor is anchored |
| $\vec{h}$ | Position of motor head |
| $\vec{F}^w$ | Force from weak bond between the motor and the microtubule |
| $\vec{a}_{\text{MT}}$ | Position on microtubule where motor is weakly bound |
| $\vec{\tau}^m$ | Torque that motor exerts on cargo |
| $\vec{c}$ | Position of cargo center |
| $\vec{\theta}$ | Cargo rotation |
| $t_n$ | Time at the $n$th time step |
| $\vec{F}^b$ | Brownian force |
| $\vec{\tau}^b$ | Brownian torque |
| $\vec{G}_c, \vec{G}_\theta$ and $\vec{G}_m$ | Gaussian random variables |

constant, $L_m$ is the motor rest length,

$$\vec{F}^w = \begin{cases} \kappa^w \left( \vec{a}_{MT} - \vec{h} \right) & \text{motor is weakly bound to the MT} \\ \\ 0 & \text{motor is unbound from the MT} \end{cases},$$

$\kappa^w$ is the weak spring between the motor head and the microtubule, and $\vec{a}_{MT}$ is where the motor head is weakly bound on the microtubule. There is a torque that is exerted on the cargo:

$$\vec{\tau}^m \left( \vec{a}, \vec{h}, \vec{c} \right) = \begin{cases} (\vec{a} - \vec{c}) \times \vec{F}^m \left( \vec{a}, \vec{h} \right) & \text{weakly bound to MT} \\ \\ 0 & \text{otherwise} \end{cases},$$

where $\vec{c}$ is the cargo center location. Thus, we have ordinary differential equations (modeled after the Langevin equation):

$$\frac{d\vec{c}(t)}{dt} = \frac{1}{6\pi\eta R} \vec{F}^m \left( \vec{a}(t), \vec{h}(t) \right) + \frac{1}{6\pi\eta R} \vec{F}^b(t), \tag{1}$$

and

$$\frac{d\vec{\theta}(t)}{dt} = \frac{1}{8\pi\eta R^3} \vec{\tau}^m \left( \vec{a}(t), \vec{h}(t), \vec{c}(t) \right) + \frac{1}{8\pi\eta R^3} \vec{\tau}^b(t), \tag{2}$$

where $\eta$ is the water viscosity, $R$ is the cargo sphere radius, $\vec{\theta}$ is the cargo orientation, and $\vec{F}^b$ and $\vec{\tau}^b$ are the Brownian force and torque, respectively, which are random variables with mean 0 and variance $2k_BT\xi_c$, and $\xi_c$ is the drag coefficient of the cargo. According to the Euler-Maruyama method, a discrete formulation of Eqs 1 and 2 will be:

$$\begin{aligned} \vec{c}(t_{n+1}) &= \vec{c}(t_n) + \frac{1}{6\pi\eta R} \vec{F}^m \left( \vec{a}(t_n), \vec{h}(t_n) \right) \triangle t \\ &+ \sqrt{2\frac{k_BT}{6\pi\eta R} \triangle t} \vec{G}_c(t_n) \end{aligned}$$

and

$$\begin{aligned} \vec{\theta}(t_{n+1}) &= \vec{\theta}(t_n) + \frac{1}{8\pi\eta R^3} \vec{\tau}^m \left( \vec{a}(t_n), \vec{h}(t_n), \vec{c}(t_n) \right) \triangle t \\ &+ \sqrt{2\frac{k_BT}{8\pi\eta R^3} \triangle t} \vec{G}_\theta(t_n), \end{aligned} \tag{3}$$

where $n$ is the current time step, and $\vec{G}_c$ and $\vec{G}_\theta$ are mutually uncorrelated vectors of independent and identically distributed (i.i.d.) Gaussian random variables with mean 0 and variance 1. The cargo cannot phase through the microtubule. Since we are simulating optical trap experiments, we add a force from the trap on the cargo:

$$\begin{aligned} \vec{c}(t_{n+1}) &= \vec{c}(t_n) + \kappa_t(\vec{c}(1) - \vec{c}(t_n)) \\ &+ \frac{1}{6\pi\eta R} \vec{F}^m \left( \vec{a}(t_n), \vec{h}(t_n) \right) \triangle t + \sqrt{2\frac{k_BT}{6\pi\eta R} \triangle t} \vec{G}_c(t_n), \end{aligned} \tag{4}$$

where $\kappa_t$ is the trap stiffness. We can now determine the motor anchor location by inputting the cargo axis of rotation $\vec{\theta}(t_{n+1}) - \vec{\theta}(t_n)$ and this axis' length as the magnitude of rotation (in

radians) into a rotation matrix $\mathbf{M}(t_n)$:

$$\vec{a}(t_{n+1}) = \mathbf{M}(t_n)(\vec{a}(t_n) - \vec{c}(t_n)) + \vec{c}(t_n) + (\vec{c}(t_{n+1}) - \vec{c}(t_n)). \tag{5}$$

Similarly to Eqs 1 and 2, we discretize ordinary differential equations for the motor head position:

$$\vec{h}(t_{n+1}) = \vec{h}(t_n) + \frac{1}{\xi_m}\vec{F}^m \triangle t + \sqrt{2D_m}\vec{G}_m, \tag{6}$$

where $\xi_m = k_B T/D_m$ is the motor drag coefficient, $D_m$ is the motor head diffusion constant, and $\vec{G}_m$ is the uncorrelated i.i.d. Gaussian random variable of mean 0 and variance 1. The motor head cannot phase through the microtubule and the cargo. Since in experiments, the microtubule lies on the coverslip surface, the motor head cannot diffuse under the microtubule.

We also considered an extended model that approximately accounts for the near-wall diffusivity hindrance from hydrodynamics [56, 57]. In lieu of fully incorporating the detailed hydrodynamics, an undertaking outside the scope of this work [55, 89], we employ the classical Brenner correction formulae [56, 58]. As an approximation based on the assumption that the $z$-directional motion dominates the binding time, we rescale the entire rotational and translation drag coefficients of the cargo by the analytical perpendicular $z$-dependent rotational and translational diffusivities. Specifically, we rescale $\eta$ in the translational update Eq (4) by $(1 - (9/8)(R/z) + (1/8)(R/z)^3)^{-1}$ and the translational $\eta$ in Eq (3) by $(1 - (5/16)(R/z)^3 + (15/256)(R/z)^6)^{-1}$ where $R$ is the radius of the cargo and $z$ is the distance from the coverslip to the center of the cargo. Notably, this neglects the asymmetry in perpendicular and parallel directions, and leave these details to future work.

Transitions between each motor state (Fig 2) are modeled as Poisson processes, with rates as follows:

$$\lambda_{\text{off}}^{\text{ADP}} = \begin{cases} k_{\text{off}}^{\text{ADP}} & \text{ADP-bound and MT-unbound} \\ k_{\text{off}}^{\text{ADP,Fast}} & \text{ADP-bound and weakly bound to MT} \\ 0 & \text{motor is ADP-unbound} \end{cases} \tag{7}$$

$$\lambda_{\text{on}}^{\text{ADP}} = \begin{cases} k_{\text{on}}^{\text{ADP}} & \text{motor is ADP-unbound} \\ 0 & \text{motor is ADP-bound} \end{cases} \tag{8}$$

$$\lambda_{\text{on}}^{\text{MT}} = \begin{cases} k_{\text{on}}^{\text{MT}} & \text{unbound from MT and within } d_{\text{MT}} \\ k_{\text{off}}^{\text{ADP,Fast}} & \text{motor is weakly bound to microtubule}, \\ 0 & \text{motor is not within } d_{\text{MT}} \end{cases} \tag{9}$$

where $d_{\text{MT}}$ is the binding distance between the motor head and the microtubule, and

$$\lambda_{\text{off}}^{\text{MT}} = \begin{cases} k_{\text{off}}^{\text{MT}} \cdot \exp F^w/F^d & \text{weakly bound to MT} \\ 0 & \text{motor is MT-unbound} \end{cases}, \tag{10}$$

where $F^d$ is the motor's critical detachment force. Parameter values are listed in Tables 1 and 3. The particular choice of $d_{\text{MT}} = 5$nm was made based on the approximate size of kinesin-1, but varying this parameter has little effect on the resulting fit (cumulative absolute fitting errors were at most 0.01s different), shown in Fig L in S1 Text.

**Table 3. Parameters used in the computational model.**

| Parameter | Description | Value |
|---|---|---|
| $\kappa^m$ | Motor stiffness (pN/nm) [12] | 0.3200 |
| $\kappa_t$ | Trap stiffness (pN/nm)† | [0.045 0.045 0.03] |
| $L_m$ | Motor length (nm)† | Varies |
| $\eta$ | Fluid Viscosity (pN · s/nm²) | 1e-05 |
| $R$ | Radius of cargo bead (nm) | 280 |
| $k_bT$ | Boltzmann constant (pN · nm) | 4.114 |
| $d_{MT}$ | MT Binding range (nm)* | 5 |
| $F^d$ | Critical detachment force (pN) [19] | 4 |

*Unmeasured estimate.

†Measured estimate.

The motor length includes the antibody that binds the motor to the cargo (about 10 nm).

**Numerical simulation.** The model is simulated forward in time. Time steps are either equal to $dt_{max}$, the maximum time step the system can undergo, or they are determined through the Gillespie-style algorithm if the next motor state-change event (i.e., bound or unbound to microtubule, ADP released or unreleased), also determined by the Gillespie-style algorithm, occurs before $t_n + dt_{max}$. An appropriate $dt_{max}$ (0.004) was chosen with a convergence test (see Fig I in S1 Text). To implement the Gillespie-style algorithm, exponential random variables from distributions with means set by each Poisson (Eqs 7, 8, 9 and 10) were generated at each time step. After the Gillespie-style algorithm determines the next event and when it occurs, the time step is used to determine the locations of the cargo center and the motor's head and anchor (Eqs 4, 5 and 6).

To mimic experimental practices, simulations are allowed to simulate 100 seconds. If the motor does not strongly bind to the microtubule during this time, the simulation starts over. This method is similar to the experiment, where the assay is run for 100 seconds screening for a binding event to occur before trapping a different cargo. The simulation is written in MATLAB, and takes approximately 0.1 seconds to simulate 1 second of the system. Example snapshots of the simulation are shown in Fig A in S1 Text.

## Model fitting, cross-validation

The model is fit through two distinct approximate inference procedures, the reconciliation of which serves as a validation for the approximations. The first procedure is a Bayesian optimization procedure [29] to obtain a single point estimate for the parameter values. The loss function is the squared distance over the mean binding times (and therefore neglects the full distributional information) and the estimated mean binding time over $S = 1000$ simulations, and these estimates are used in Fig 3 and Table 1.

To obtain uncertainty quantification seen in Fig 4, we also employ a sequential Monte Carlo approximate Bayesian computation approach [30]. These techniques are far slower than the optimization procedure and require the specification of a prior distribution for each parameter, but provide some notion of uncertainty quantification, and were used to generate Fig 4 with some data withheld. That is, because of the heavy computation expense, only the shortest motor at 0 nm average distance between the cargo and microtubule, the mid-length motor at 40 nm average distance, and the longest motor at 80 nm average distance were used in the fitting. The maximum a posteriori (MAP) estimates from this latter procedure closely

**Table 4. Hyperparameters for priors used in estimated posterior densities, all taken to be lognormal distributions.**

| Parameter | Mean | Standard Deviation |
|---|---|---|
| $k_{\text{off}}^{\text{ADP}}$ | $10^{-2}$ | 0.001 |
| $k_{\text{on}}^{\text{ADP}}$ | $10^{3}$ | 100 |
| $k_{\text{off}}^{\text{ADP,Fast}}$ | $10^{0.3}$ | 1 |
| $k_{\text{on}}^{\text{MT}}$ | $10^{1.7}$ | 100 |
| $k_{\text{off}}^{\text{MT}}$ | $10^{-1}$ | 0.01 |
| $D_m$ | $10^{3.3}$ | 100 |
| $\kappa_w$ | $10^{-2.7}$ | 0.001 |

agree those of the first procedure, supporting the validity of both. Furthermore, in Table A in S1 Text, we show the inference procedures successfully infer rates from synthetic data. Lognormal priors are chosen for all parameters, and hyperparameters are shown in Table 4. Hyperparameters were chosen based on the range of values reported in the literature for each parameter when available. Otherwise, they were chosen to be approximately uninformative with large standard deviations. Initially, 100 simulations estimate the binding times in the model, and weights in the sequential Monte Carlo algorithm are defined as $w_i = \pi/(\sum w_{i-1}K_i)$, where $K_i$ is the perturbation kernel for the $i$th sequence, $i > 1$. We use a Gaussian distribution for $K$. These new samples are then used to simulate more mean binding times until 100 samples are generated resulting in a relative error lower than 1.8. Eight more sequences follow in this same manner, each time the relative error threshold decreases by 0.2. A kernel density estimator was then applied to the resulting samples shown in Fig 4.

The cross-validation procedure in Fig 3 was implemented by fitting the models using the aforementioned point estimate optimization scheme with data withheld, and then test error defined to be $N^{-1}\sum_{i=1}^{N}(t_i - \hat{t}_i)/t_i$, a percentage error over the test scenarios. This procedure is validated in Fig J in S1 Text showing cross-validation successfully identifying the correct model when tested against synthetic data.

## Supporting information

**S1 Text. Supporting figures and tables.** Additional demonstrations, investigations, and validations of computations described in the main text.
(PDF)

## Acknowledgments

We thank Brennan Sprinkle for helpful discussions about Brownian hydrodynamics. Fig 2 includes an icon (lipid-hdl-1 icon) by Servier https://smart.servier.com/, which is licensed under CC-BY 3.0 Unported. This work utilized the infrastructure for high-performance and high-throughput computing, research data storage and analysis, and scientific software tool integration built, operated, and updated by the Research Cyberinfrastructure Center (RCIC) at the University of California, Irvine (UCI). The RCIC provides cluster-based systems, application software, and scalable storage to directly support the UCI research community. https://rcic.uci.edu.

## Author Contributions

**Conceptualization:** Steven P. Gross, Christopher E. Miles.

**Data curation:** Babu Reddy Janakaloti Narayanareddy.

**Formal analysis:** Trini Nguyen, Babu Reddy Janakaloti Narayanareddy.

**Funding acquisition:** Trini Nguyen, Steven P. Gross.

**Investigation:** Trini Nguyen, Babu Reddy Janakaloti Narayanareddy, Steven P. Gross.

**Methodology:** Trini Nguyen, Babu Reddy Janakaloti Narayanareddy, Steven P. Gross, Christopher E. Miles.

**Project administration:** Steven P. Gross, Christopher E. Miles.

**Resources:** Babu Reddy Janakaloti Narayanareddy.

**Software:** Trini Nguyen, Babu Reddy Janakaloti Narayanareddy.

**Supervision:** Steven P. Gross, Christopher E. Miles.

**Validation:** Trini Nguyen.

**Visualization:** Trini Nguyen, Christopher E. Miles.

**Writing – original draft:** Trini Nguyen.

**Writing – review & editing:** Trini Nguyen, Babu Reddy Janakaloti Narayanareddy, Steven P. Gross, Christopher E. Miles.

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
