## [Decision Letter · Decision Letter 0]

16 Apr 2024

Dear Miles,

Thank you very much for submitting your manuscript "Competition between physical search and a weak-to-strong transition rate-limits kinesin binding times" for consideration at PLOS Computational Biology. As with all papers reviewed by the journal, your manuscript was reviewed by members of the editorial board and by several independent reviewers. The reviewers appreciated the attention to an important topic. Based on the reviews, we are likely to accept this manuscript for publication, providing that you modify the manuscript according to the review recommendations.

Sincerely,

Changbong Hyeon

Academic Editor

PLOS Computational Biology

Daniel Beard

Section Editor

PLOS Computational Biology

Reviewer's Responses to Questions

**Comments to the Authors:**

Reviewer #1: This is a worthy contribution that combined results from optical trapping and simulations to estimate the time it takes a kinesin motor on a spherical cargo to bind to a microtubule. Others have focused on the rotational diffusion of a cargo and how the motor itself diffuses around on the cargo. In this paper an additional rate-limiting step (ADP release from motor) is proposed.

This information, although done here for a very simplified system, could be useful to understand when and how a vesicle with motors will actually start moving. The manuscript is written well and logically organized, I found it easy to follow. I recommend publication with some changes. I have some major comments and several minor ones :-

Major Comments :-

The ADP release step seems to slow down the kinesin-microtubule binding by ~1 second when the binding is most favorable (cargo is very close to microtubule; Fig 3). What does this number (~1 second) really mean for cargo inside a cell which could have several kinesin and dynein motors close to the microtubule ? Earlier papers from the Gross group (and also from other groups) have brought up the possibility that kinesin is outnumbered by dynein motors on some cellular cargos. Given the On/Off rates for dynein, what are the implications of the present findings to a cargo which has both kinds of motors ? There should be some discussion on this aspect.

Section "Broader Lessons for Probing Protein-Protein Interactions"

The discussion here is not really broad. It should go beyond motor-microtubule binding (considering the way the Introduction to the paper is written). I strongly feel that situations outside this (motor-microtubule) example should be discussed. How often, and where else is an ADP-dependent rate relevant? What about prot-prot interactions where ADP is not a factor?

Minor comments :-

"of diffuse importance across cellular function"

... conveys a negative outlook to the paper. What are the authors trying to say? Should diffuse be re-worded?

"After this initial binding, other motors associated with the cargo attach to the filament and go through

cycles of reattachment."

Most kinesin binding rates are known from in vitro bead-kinesin assays where kinesin is very dilute and it is largely a single kinesin-to-microtubule interaction. The point that authors make is valid in a more in vivo context (many motor), but perhaps not so much to the presently known measures of binding times done in vitro. So, relevance of this statement in the context of this study is unclear.

"detectable direction of the bead"

detectable direction of the bead

OR

detectable motion of the bead ?

Simple diffusion model appears closer to experimental data in Fig3C. Then why is the average percent error higher for diffusion model in Fig 3D?

Lines 113-114

You should mention the magnitude of this delay in the text here for the reader to appreciate better (delay looks like ~1 sec from Fig 3a)

Reviewer #2: In this manuscript, the authors study the binding process of a kinesin motor to the microtubule. Because the binding process can be defined in many different ways, for example including diffusion of the motor with its cargo close to the microtubule or other geometric aspects, the authors choose to study a well-defined system that has been experimentally realized: a single motor attached to a bead held in an optical trap at a defined height above the microtubule. When the motor pulls the bead against the trap’s restoring force, it will unbind and eventually rebind to the microtubule. This time between the unbinding and the rebinding event is conceptualized by different mechanistic models fitted to experimental data. From this modeling approach, the authors conclude that a model that includes an ADP-state dependence of the motor head is more consistent with the experimental data than a model based only on a diffusive search.

The authors use simulation-based inference methods to fit a stochastic simulation of a coarse-grained model to the experimental data. Overall, using such advanced statistical methods to combine stochastic models with experimental data is an important step forward in the field of computational biology. The presentation of the data and the logic in the text is very clear. The conclusions are solid and therefore I recommend publication in PLOS CB after the authors have addressed my comments which I suspect will improve the quality of the paper.

1) As presented in the manuscript, the bead is always kept at a specific height above the MT by a feedback trap. This scenario is of course very artificial and not very likely to occur in cells or in vitro experiments. I am wondering if it would not have been possible to switch off the feedback once the motor has been unbound to mimic the scenario of a bead starting its binding process at a defined height above the MT. I would suggest that the others could use their simulation to find out what will happen to the binding process if the bead is not fixed at a certain height.

2) Although the authors claim that they work under single-motor conditions in their experiments, they do not show any dilution curve, therefore there is still the possibility of multi-motor conditions.

3) Fig.1: It’s not clear why the authors labeled the fourth cartoon “walking outside trap”. The motor is not walking outside the trap, it is walking away from the trap center. I don’t know if the authors ever see a bead that is pulled outside of the laser beam.

4) Fig 3: According to the text, panel C is only a validation plot. This could be indicated in the title of the plot. Is there a reason why the D+A model systematically overestimates the binding time for the 60 nm motor?

5) In the caption of Fig. 4, the authors say “approximate Bayesian Computation (ABC), but in the text, they talk about “sequential approximate Bayesian Computation (sABC).”, which could be confusing. I think it would be nice if the authors could also include in Table 1 the parameter estimates from the sABC, extracted from the posteriors of Fig 4. I don’t know the best practice. Maybe to report the mean or the median of the posteriors? Furthermore, it is interesting to see that the priors for the kADP,off, and kMT,on are much more restrictive than for the other parameters, was there a reason for this?

6) I think the effective description of the model needs clarification. It is not clear at all what times are measured and what is shown in fig. 5A. In line 204 it says: “fraction of time stochastic simulations of the full binding process spent in State 1 and 2,...” The title of the plot says: ”Proportion of time spent in states 1 or 2”. I think the “or” should be an “and”?. Is proportion always with respect to the total binding time,? How do the authors calculate these proportions? Let’s say the mean time in state 1 is <t_1>, the mean time in state 2 is <t_2>, and the total time is <t_tot>. Is the proportion that they plot: <t_1> + <t_2>/<t_tot> or do they first determine the proportion and then average, like <(t_1 + t_2)/t_tot>? Panel C: I find it very confusing that the rates that are shown in this panel kADP,off and so on are not the same rates (numerical values) as in Fig.2 Those rates are somehow effective rates and therefore should also get different symbols.

7) Line 223 “Figand” -> “Figure”

8) I understand that the authors cannot exclude all possible models and can only show that their model is consistent with the data. This point was very well addressed in the Discussion. However, one possibility that the authors mention in line 344 is the well-known result of the distance-dependent diffusion coefficient of an object close to a surface. The authors could use the Brenner formula to estimate this effect.

9) I was surprised to see that the authors do not discuss a potential experiment changing the [ADP] concentration. The [ADP] concentration would change the rates in the system and thus indicate if the system is indeed influenced by the ADP state of the motor. Unfortunately, it was also not clear what the [ADP] concentration was in the experiment and if an ATP regeneration buffer was used.

10) In the supplements, the authors should consistently use SI units: it is neither (secs), nor (sec), it is only (s) for units in seconds. Fig S5. The rates should get symbols and the average rates used for the results of Fig.5 should be listed somewhere.</t_tot></t_2></t_1></t_tot></t_2></t_1>

Reviewer #3: The authors develop a simple Brownian dynamics simulation to study the initial attachment kinetics of kinesin to the microtubule (MT). They consider two mechanisms, one where motor attachment is limited by diffusion only, versus the other where ADP release following diffusional search is rate-limiting. They find the latter to better agree with experimental data. They make careful arguments about validation and limitations of their model. This is a potentially useful contribution to understanding the behavior of the kinesin motor that may also find applications in examining the attachment kinetics of other motors to their track. However, there are several points that need clarification and elaboration:

- Fig 3D: Average Percent Test Errors for 33nm and 60nm are over 50% for both mechanisms, which I think cannot be used to support that the better fit by the D+A model (diffusion and ADP release) is not just because it has more parameters.

- Line 425: Typo "K-560-His" should be "DK-560-His"

- It will be very helpful to define various vectors and quantities introduced in pp15-17 in a figure, either Fig 2, SI Fig 1, or in a new figure.

- Line 478: Statement here indicates that their model is 3-dimensional (3D). But SI Fig 1 seems to indicate this is a 2D model. If this is indeed a 3D model, an explanation of how the orientation of the cargo (theta; line 509) is assigned and handled in Eq 2, as Langevin equation for the rotational diffusion is more involved, often requiring the use of quaternion.

- Clarify different versions of theta: in Eq 2, it has a tilde, on line 509, it is a scalar, and in places below, a vector symbol is used.

- How does the simulation handle cases when the cargo diffuses away from the MT?

- Table 2: They use a fixed 5nm as the MT binding range (d_MT). If d_MT is shorter, k_on^MT will decrease, making diffusional encounter to be more rate-limiting. I suggest to test a few more values of d_MT to check how sensitive their fit parameters vary (especially k_off^{ADP,fast} vs k_on^MT).

- Their experimental buffer contained ATP (line 434). Explain what will happen when no ATP is present. According to their model, the initial binding should not be affected because ATP is not needed in this process.

**Have the authors made all data and (if applicable) computational code underlying the findings in their manuscript fully available?**

Reviewer #1: Yes

Reviewer #2: Yes

Reviewer #3: Yes

PLOS authors have the option to publish the peer review history of their article (what does this mean?). If published, this will include your full peer review and any attached files.

Reviewer #1: No

Reviewer #2: No

Reviewer #3: No

Figure Files:

Data Requirements:

Please note that, as a condition of publication, PLOS

---

## [Editor Report · Decision Letter 1]

10 May 2024

Dear Miles,

We are pleased to inform you that your manuscript 'Competition between physical search and a weak-to-strong transition rate-limits kinesin binding times' has been provisionally accepted for publication in PLOS Computational Biology.

Best regards,

Changbong Hyeon

Academic Editor

PLOS Computational Biology

Daniel Beard

Section Editor

PLOS Computational Biology

---

## [Editor Report · Acceptance letter]

14 May 2024

PCOMPBIOL-D-24-00454R1 

Competition between physical search and a weak-to-strong transition rate-limits kinesin binding times

Dear Dr Miles,

I am pleased to inform you that your manuscript has been formally accepted for publication in PLOS Computational Biology. Your manuscript is now with our production department and you will be notified of the publication date in due course.

With kind regards,

Zsofia Freund
